# Psychometric properties of a modified health belief model for cervical cancer and visual inspection with acetic acid among healthcare professionals in Ethiopia

Semarya Berhe Lemlem[1☯*], Rebecca A. Gary[2☯], Katherine A. Yeager[2☯], Mitike Molla Sisay[3☯], Melinda K. Higgins[2☯]

1 Department of Midwifery, School of Nursing & Midwifery, College of Health Sciences, Addis Ababa University, Addis Ababa, Ethiopia, 2 Nell Hodgson Woodruff School of Nursing, Emory University, Atlanta, Georgia, United States of America, 3 School of Public Health, College of Health Sciences, Addis Ababa University, Addis Ababa, Ethiopia

☯ These authors contributed equally to this work.
* semitaye@yahoo.com

**Data Availability Statement:** The data set used for this study could be traced with the following DOI

## Abstract

### Purpose

Evidence supports that the Health Belief Model (HBM) can explain and predicts certain health behaviors, including participation in cervical cancer (CC) screening. The purpose of this study was to evaluate the psychometric properties of a modified HBM for CC and visual inspection with acetic acid (VIA) in female healthcare professionals in Addis Ababa, Ethiopia, 2020.

### Methods

Psychometric properties related to CC and VIA were tested using 42-item modified HBM self-administered questionnaire and a cross-sectional study design with simple random sampling. Kaiser-Meyer-Olkin and Bartlett's sphericity test indicated that data sampling adequacy for exploratory factor analysis was 0.792 ($\chi 2$ = 3189.95, df = 351, p < .001). Items with cross-loading and factor loadings $\geq$ 0.5 were retained. Confirmatory factor analysis (CFA) was conducted to determine model fit.

### Results

The final analysis included 194 women, (mean age 30±4.34). Twelve items with $\leq$ 0.5 were removed and 30 retained items loaded into 6 factors; (benefits of VIA, perceived seriousness of CC, barrier (fear of negative outcome), self-efficacy, susceptibility to CC, and barriers (health system delivery)) explained 65% of the total variance. Cronbach's alpha for the total instrument was 0.8 and reliability for the 6 subscales was 0.76–0.92. Composite reliability and average variance extracted indicated good internal consistency and convergent validity. CFA identified 6 additional items to be removed with high residual covariance. The final 24 items of the modified HBM had an acceptable model fit (goodness-of-fit index (GFI)

number: https://doi.org/10.20372/aau_rdr/
GXJ5UR.

**Funding:** Addis Ababa University and The office of the Director for Research in collaboration with Swedish International Development Cooperation Agency (SIDA) Grant coordination, Addis Ababa University women researchers funded this work. The funders had no role in study design, data collection and analysis, decision to publish, or preparation of the manuscript. The correct grant numbers for the awards received for our study is RD/LT-159/2019 by the office of the Director for Research in collaboration with SIDA Grant coordination called AAU women researchers.

**Competing interests:** The authors have declared that no competing interests exist.

= 0.861, adjusted GFI = 0.823, comparative fit index = 0.937, root mean square error of approximation = 0.059).

## Conclusion

The modified HBM for CC and VIA with 24 items had adequate psychometric properties and may be used by Ethiopian healthcare professionals for research or clinical purposes. To support external validity the updated 24 items tool is suggested for application in further study in different populations in Ethiopia.

## Introduction

The global burden of cervical cancer is projected to continue to increase, rising to approximately 700,000 diagnosed, with an estimated 400,000 deaths in 2030. Most of these increases will be among women in low and middle income countries (LMICs), reflecting the severity of the global divide in cervical cancer morbidity and mortality [1, 2]. In LMICs, including Ethiopia, cervical cancer the leading cause of death. In 2010, the Ethiopian Ministry of Health reported that there were 4,648 new cases and 3,235 deaths due to cervical cancer, equating to a mortality rate of 18.4/10000 [3–5].

Screening is the single most important public health strategy to reduce cervical cancer incidence and consequent mortality [6, 7]. Survival rates for cervical cancer in Sub Saharan Africa are 21% compared with 70% in the USA and 66% in Europe [8, 9]. Despite the negative clinical outcomes associated with cervical cancer in Ethiopia, approximately 27.19 million women are estimated to be at risk, yet less than 1% of women aged 18–69 years undergo screening every 3 years, as recommended by the World Health Organization (WHO, 2014) and adopted by Ethiopia [10–12]. Effective, low resource screenings and treatment methods are recommended in the WHO guideline, which includes use of "see and treat" screening strategy with visual inspection with acetic acid (VIA) as the primary screening method and cryotherapy as a treatment option [4].

Research to date indicates that the availability of screening services is inadequate to increase screening participation [10, 13, 14]. Further some behaviors and beliefs may significantly impact the decisions of women to take preventive actions against cervical cancer [15]. Much of the work associated with cancer screening has been informed by the Health Belief Model (HBM). The HBM focuses on preventing illness occurrence by encouraging health behaviors that avoid disease [15]. Researchers have reported that diverse demographic, psychosocial, and health beliefs may influence cervical screening perceptions and, thus, indirectly influence health-related behavior [16]. To improve screening participation among women, a better understanding of their health beliefs is essential [17].

There is considerable empirical support that HBM can explain and predict certain health behaviors, particularly cancer screening [16]. For behavior change to occur such as obtaining cervical cancer screening, people must perceive a threat from their current behavioral patterns (perceived susceptibility and severity) and believe that change of a specific behavior will result in a valued outcome at an acceptable cost (perceived benefit). Individuals also must feel competent (self-efficacy) to overcome perceived barriers to taking action [18].

One of the most important limitations in both descriptive and intervention research regarding HBM has been variability in the measurement of central HBM constructs. Construct definitions need to be consistent with HBM theory as originally conceptualized, and measures need to be specific to the behavior being addressed and relevant to the population among

whom they will be applied [19]. The reliability of instruments may differ among populations for various reasons, including socio-demographic factors and cultural nuances.

Testing a modified version of the HBM scale modified for cervical cancer and VIA among women in Ethiopia will contribute to and expand knowledge in this area. A survey conducted among Ethiopian healthcare professionals revealed a significant deficit regarding cervical cancer, which could have implications for future screening programs since these providers would likely play a principal role in patient education and implementation of a cervical cancer screening program in Ethiopia [18]. Cervical screening behaviors among healthcare professionals working at the College of Health Sciences in Ethiopia have not previously been examined using a modified version of Champion's HBM. Therefore, the objective of this study was to evaluate and validate the psychometric properties of the modified HBM scale for Cervical Cancer and VIA (42) questionnaire in female Ethiopian healthcare professionals.

## Methods

### Study design and settings

A cross-sectional study design was used to test the psychometric properties of the modified HBM for cervical cancer and VIA for application in female healthcare professionals working at the College of Health Sciences at Addis Ababa University, Ethiopia in 2020.

### Sample size

The sampling frame for this study was based on participants' profile obtained from the office of human resource. Simple random sampling was used to select the participants' who fulfills the inclusion criteria being female healthcare professionals aged 21–65 years, no prior history of cervical cancer, speaks English and employed full-time. For this survey students and non-health care professionals were excluded from the study. Samples were proportionally allocated based on the number of female staffs available in the units/ departments. Health care professionals who fulfil the inclusion criteria and willing to participate were approached face to face to fill out the semi structured questionnaires. For psychometric testing, there are a wide range of recommendations for adequately powered analysis ranging from 3 participants to 1 item to 5:1 [19]. In the current study a 5:1 ratio of participants to items was used to perform exploratory factor analysis (EFA). The modified HBM scale to be tested included 42 items, bringing the required number of participants to 210. Given the 24 items in the final modified version presented here the 5:1 ratio increased to 8.75:1.

The study received ethical approval from the Institutional Review Board of Addis Ababa University (protocol number, 017/20/-Nursing) before its initiation. Written informed consent was collected from all participants. All potential participants were informed about the purpose, method and potential benefits and knowledge gained from the study. Data for the psychometric evaluation were collected from June to August 2020.

### Instruments

A structured questionnaire was used to collect socio-demographic data including age, marital status, work experience, profession, income and level of education. In addition, questions about prior cervical cancer screening practice were included.

### The modified HBM scale for cervical cancer and VIA use

The HBM questionnaire used in the current study was modified from the HBM Scale for Cervical Cancer and Pap Smear Test reported by Guvenac (2011), which had been previously

modified from Champion's HBM by inclusion of four additional question to the barrier construct [20]. The self-efficacy construct was not tested by Guvenac (2011) and was considered to be a modified HBM scale. The following HBM constructs were included: susceptibility, seriousness, benefits/ motivation, barriers, health motivation, and self-efficacy. The modified HBM scale was further adapted for this study to refer to cervical cancer screening as VIA since the Pap smear is not widely used in Ethiopia. Visual inspection with acetic acid (VIA) is visualization of woman's cervix to detect precursors of cervical cancer after application of acetic acid (ordinary table vinegar) on her cervix and it is a simple, low-cost, and efficient alternative to cytologic testing in low-resource areas [21]. The Health Belief Model Scale for Cervical cancer and VIA has 42 items and six subscales, including benefit (8 items), barrier (18 items), seriousness (7 items), susceptibility (3 items), health motivation (3items), and for self-efficacy (3 items). All subscale items have the following five-point Likert-type response choices: strongly disagree (1 point), disagree (2 points), neutral (3 points), agree (4 points), and strongly agree (5 points). Permission to test the modified instrument and to adapt the scale for use by women in Ethiopia was obtained from Guvenac.

## Data analysis and presentation

Data were screened for missing and outlier values and data entry errors using the frequency distributions of the variables and by inspection of entered data. Data were exported to SPSS version 25 software for analysis.

Construct validity of the scale was examined using EFA and principal axis factoring (PAF) extraction with oblimin rotation. The loading criterion was set as $\leq 0.3$ and Bartlett's tests was used to analyze sampling adequacy. Reliability included internal consistency and was estimated using Cronbach alpha coefficient values for the different domains of the instrument. Cronbach's alpha between 0.70–0.90 is considered to reflect adequate internal consistency [22].

The Kaiser-Meyer-Olkin (KMO) measure of sampling adequacy was used to quantify sufficiency of item correlation for performing factor analysis. The KMO index ranges from zero to one and, the minimum acceptable value is 0.60. Bartlett test of sphericity test was applied to assess the presence of correlations among variables significant value ($p < 0.05$) indicates the appropriateness of EFA [23]. Generally, the results indicated strong factorability and supported conducting an EFA.

Factor extraction analysis was conducted using PAF to determine the number of factors to retain, as follows: (a) using the Guttmann- Kaiser greater-than-one rule which recommends that only those factors with eigenvalues $\geq 1.0$ be retained; (b) using the Cattell scree test, which involves constructing 16 plots of extracted factors against their eigenvalues in descending order of magnitude; (c) by applying the Monte Carlo Parallel analysis rule, which compares factor eigenvalues to a set of eigenvalues generated from random data, and recommends retaining those eigenvalues that exceed the corresponding values from the random data; and, (d) by calculating the amount of total variance explained by the communality of each variable to determine the number of factors to be preserved [24]. PAF was conducted together with oblique rotation (direct oblimin) which is often seen as producing more accurate results for research involving human behaviors. Regardless of which rotation method is used, the main objectives are to provide easier interpretation of results, and produce a solution that is more parsimonious [25] Items were deleted from the EFA if factor load was $< 0.5$, loaded on more than one factor, or not loaded on any factor [25].

In confirmatory factor analysis (CFA), model fitness was assessed by (a) over model fitness, using the Chi test $\chi^2$ (p) ($p > 0.05$) and normed (Chi-square and df ratio) $\chi^2$ (CMIN/df) $\leq 3$;

(b) incremental fitness with comparative fit index (CFI) and normed fit index (NFI) $\geq$ 0.95 and; (c) absolute fitness with, goodness-of-fit index (GFI), adjusted GFI (AGFI) $\geq$ 0.80, root mean square residual (RMR) $\leq$ 0.08, and root mean square error of approximation (RMSEA) $\leq$ 0.06 [26]. The method used to potentially further refine /remove items from the instrument was with load <0.3 or, higher covariance, and estimates of the residual variance.

The criteria for convergent validity were as follows composite reliability $\geq$ ± 1.97 (p < 0.05); and average variance extracted (AVE) $\geq$ 0.50 [27].

Participant's demographic characteristics were summarized using descriptive statistics. Categorical variables are presented as percentages and frequencies.

### Inclusivity in global research

Additional information regarding the ethical, cultural, and scientific considerations specific to inclusivity in global research is included in the Supporting Information (S1 Checklist).

## Results

### Socio-demographic characteristics

We approached 210 female participants at the College of Health Sciences. Variables were checked for outliers based on the Mahalanobis distance and 16 cases with p< 0.001 were excluded from the analysis resulting a final study sample of 194 female healthcare professionals (response rate 92.4%) working at Addis Ababa University College of Health Sciences, with mean age 33.14$^\pm$7.2 years. The majority of participants 150 (77.3%) worked in the clinics and 102 (52.6%) had < 5years of work experience. The socio-demographic characteristics of participants are presented in Table 1.

### Participant screening practice

The majority of the women healthcare professionals 138 (71.1%) enrolled in the study had not been previously screened for cervical cancer in their life time. Only 33 (17.0%) of the sample had been screened for cervical cancer in the past three years and the remaining 161(83.0%) had not.

### Validity and reliability

The KMO and Barlett tests were conducted done before the EFA. KMO should be 0.6 to continue with factor analysis. In more detail, KMO values with 0.90 are considered excellent; 0.80 good, 0.70 middle range; 0.60 mediocre; 0.50 acceptable; and $\leq$ 0.50 is unacceptable [28]. The KMO measurement of sampling adequacy was 0.792, and the Bartlett's test of sphericity was significant ($\chi$2 = 3189.95, df = 861, p < .001) indicating the adequacy of the sample (n = 194) for EFA.

The Cronbach's alpha value for total items was 0.80 and the reliability coefficient values calculated for each sub dimension were 0.92 for benefit of VIA, 0.87 for perceived seriousness of cervical cancer, 0.87 for barrier (fear of negative outcome), 0.82 for self-efficacy, 0.76 for susceptibility to cervical cancer, and 0.80 for barriers (health system delivery).

### Exploratory factor analysis

Decisions regarding the number of extractable factors included in EFA were made using eigenvalues, factor loadings, and scree plot diagrams. Initial PAF showed the presence of nine factors with eigenvalues > 1. A review of the scree plot revealed a break before four factors. Supporting Catelli's scree test, Monete Carlo parallel analysis showed seven factors with

**Table 1. Socio-demographic characteristics of health care professionals working in College of Health Sciences Addis Ababa University (n = 194).**

| Variables | Category | Frequency | Percent % |
|---|---|---|---|
| Age | < 30 Years | 101 | 52.1 |
| | 30–34 years | 40 | 20.6 |
| | 35–39 Years | 23 | 11.9 |
| | ≥ 40 Years | 30 | 15.5 |
| Marital Status | Single | 76 | 39.2 |
| | Married | 108 | 55.7 |
| | Others | 10 | 5.2 |
| Service Year | <5 years | 102 | 52.6 |
| | 5–10 Years | 51 | 26.3 |
| | >10 years | 41 | 21.1 |
| Educational level | BSc | 124 | 63.9 |
| | Masters | 44 | 22.7 |
| | MD | 20 | 10.3 |
| | Others* | 6 | 3.1 |
| Monthly income (ETB) | ≤6193 | 77 | 39.7 |
| | 6194–9056 | 71 | 36.6 |
| | >9056 | 46 | 23.7 |
| Professional stream | Clinical | 150 | 77.3 |
| | Academic | 39 | 20.1 |
| | Both | 5 | 2.6 |
| Professional title | Nurse | 128 | 66.0 |
| | Midwife | 18 | 9.3 |
| | Physicians | 22 | 11.3 |
| | Others** | 26 | 13.4 |
| Unit of work | Medical ward | 29 | 14.9 |
| | Surgical ward | 22 | 11.3 |
| | Oncology | 21 | 10.8 |
| | Outpatient | 44 | 22.7 |
| | Others*** | 78 | 40.2 |

**Note:** Others: Widowed and divorced;

*: Diploma and PhD;

**, anesthetist, pharmacy, laboratory, radiology;

*** EPI, FP, etc

eigenvalues surpassing the corresponding criterion values for a randomly generated data matrix of the same size for ample EFA. Parallel analysis is a consistent and acceptable method used to precisely decide the number of factors [28]. Of 42 items30 were retained and total of twelve items were eliminated based on factor loading lower than 0.5, cross-loading, or a communality value < 0.40.

A key component analysis of the scale revealed that six factors had values > 1. Of the factors revealed by oblimin rotation, the first one explained 19.01% of the total variance, while the second to sixth explained 12.594%, 10.018%, 6.526%, 4.789%, and 4.424% respectively; the variance explained by all six factors was 65%. Factor analysis identified that the first factor, benefits of VIA included 8 items; the second factor perceived seriousness of cervical cancer had 6 items; the third barrier to negative health outcome had 3 items, the fourth self- efficacy had 5 items; the fifth susceptibility to cervical cancer had 3 items, the sixth barriers to health

delivery had 5 items. Items in the health motivation subscale changed sub scale and were loaded on the self-efficacy sub scale, and the barrier sub scale was divided into two factors, while the remaining items remained in their original subscales. Factor load values of the items in the first to sixth factors were: 0.635 to 0.834, while it was 0.602–0.857, 0.571–0.956, 0.553–0.835, 0.585–0.900, and 0.576–0.772 respectively in Table 2.

To check internal consistency Cronbach's alpha and composite reliability coefficients calculated for the six factors. In addition, AVE values were estimated to assess convergent validity. Both alpha and composite reliability $\geq 0.70$ and AVE $\geq 0.5$ reflect adequate internal consistency and convergent validity. Cronbach's alpha, composite reliability, and AVE values were calculated for each factor, and good values were obtained for all subscales in Table 3.

## Confirmatory factor analysis

Based on EFA conducted out in the initial phase, a six-factor model of the modified HBM and VIA questionnaire was evaluated using a randomly allocated sample (N = 194) by CFA.

**Table 2. Factor loading for EFA with oblimin rotation modified HBM for VIA-30 among health care professionals, Ethiopia 2022.**

| ITEM | BEN | SER | BAR 1 | EFF | SUS | BAR 2 |
|---|---|---|---|---|---|---|
| 6 Maintaining good health is extremely important to me | .834 | | | | | |
| 1 Having regular VIA will help to find changes to the cervix, before they turn into cancer | .826 | | | | | |
| 5 I want to discover health problems early | .815 | | | | | |
| 8 I feel it is important to carry out activities which will improve my health | .797 | | | | | |
| 3 I think that having a regular VIA is the best way for cervical cancer to be diagnosed early | .757 | | | | | |
| 7 I look for new information to improve my health | .732 | | | | | |
| 4 Having regular VIA will decrease my chances of dying from cervical cancer | .720 | | | | | |
| 2 If cervical cancer was found at a regular VIA test its treatment would not be so bad | .635 | | | | | |
| 29 I am afraid to think about cervical cancer | | .857 | | | | |
| 27 The thought of cervical cancer scares me | | .802 | | | | |
| 28 When I think about cervical cancer, my heart beats faster | | .799 | | | | |
| 32 If I had cervical cancer my whole life would change | | .676 | | | | |
| 30 Problems I would experience with cervical cancer would last a long time | | .621 | | | | |
| 31 Cervical cancers would threaten a relationship with my boyfriend, husband, or partner | | .602 | | | | |
| 10 I am afraid to have a because I don't know what will happen | | | .956 | | | |
| 9 I am afraid to have VIA for fear of a bad result | | | .954 | | | |
| 12 I would be ashamed to lie on a gynecologic examination table and show my private parts to have a VIA | | | .571 | | | |
| 41 I feel capable of getting a VIA test. | | | | .835 | | |
| 42 I feel capable of managing any emotional distress caused by VIA test | | | | .792 | | |
| 40 I feel capable of arranging to have a VIA test. | | | | .641 | | |
| 38 I exercise at least 3 times a week for my health | | | | .603 | | |
| 39 I have regular health check-ups even when I am not sick | | | | .553 | | |
| 36 I feel I will get cervical cancer sometime during my life | | | | | .900 | |
| 34 It is likely that I will get cervical cancer in the future | | | | | .680 | |
| 35 My chances of getting cervical cancer in the next few years are high | | | | | .585 | |
| 25 My partner does not want me to get VIA | | | | | | .772 |
| 26 It is difficult to get an appointment for VIA | | | | | | .751 |
| 22 I will never have a VIA if I have to pay for it | | | | | | .630 |
| 24 The VIA may move the intrauterine device | | | | | | .577 |
| 23 I do not have time to get VIA | | | | | | .576 |

Note, n = 194. Factor loadings > .5 only presented. BEN: Benefit of VIA, SER: Perceived seriousness of cervical cancer, BAR 1: Barrier fear of negative outcome, EFF: Self-efficacy, SUS: Susceptibility to cervical cancer, BAR 2: Barrier to health delivery

**Table 3. Cronbach's alpha, composite reliability and AVE for each factor of modified HBM for cervical cancer and VIA among health care professionals, Ethiopia.**

| Factor name | No of items | α Cronbach | CR | AVE |
|---|---|---|---|---|
| **30 items (item11,13,14,15,16,17,18,19,20,21,33,37 removed)** | | | | |
| Benefits of VIA | 8 | 0.92 | 0.92 | 0.59 |
| Perceived Seriousness of cervical cancer | 6 | 0.87 | 0.87 | 0.54 |
| Barrier Fear of negative outcome | 3 | 0.87 | 0.88 | 0.72 |
| Self-efficacy | 5 | 0.82 | 0.82 | 0.48 |
| Susceptibility to cervical cancer | 3 | 0.76 | 0.77 | 0.54 |
| Barrier Health delivery | 5 | 0.80 | 0.80 | 0.44 |
| **26 items (item7, 5, 39 and 22 removed)** | | | | |
| Benefits of VIA | 7 | 0.90 | 0.91 | 0.58 |
| Perceived Seriousness of cervical cancer | 5 | 0.86 | 0.86 | 0.56 |
| Barrier Fear of negative outcome | 3 | 0.86 | 0.88 | 0.72 |
| Self-efficacy | 4 | 0.82 | 0.83 | 0.56 |
| Susceptibility to cervical cancer | 3 | 0.76 | 0.78 | 0.55 |
| Barrier Health delivery | 4 | 0.75 | 0.78 | 0.48 |
| **24 items (item23 and 38 removed)** | | | | |
| Benefits of VIA | 7 | 0.90 | 0.91 | 0.58 |
| Perceived Seriousness of cervical cancer | 5 | 0.86 | 0.86 | 0.56 |
| Barrier Fear of negative outcome | 3 | 0.86 | 0.88 | 0.72 |
| Self-efficacy | 3 | 0.88 | 0.88 | 0.71 |
| Susceptibility to cervical cancer | 3 | 0.76 | 0.77 | 0.54 |
| Barrier Health delivery | 3 | 0.79 | 0.79 | 0.56 |

Note: (N = 194), CR, composite reliability, AVE Average Variance Extracted

conducting confirmatory factor analysis with 30 items, which offered valuable evidence about scale stability. We removed a total of 4 items that showed higher covariance in the standardized residual covariance matrices during CFA; item 1 from the perceived benefit factor, item 22 from perceived barrier towards health delivery, item 30 from perceived seriousness and item 39 from perceived self-efficacy. The factor scores for each construct were positively correlated with one another. Each item of the construct was correlated expressively at $\geq 0.5$ except for two items (item 23 and 38). EFA was run again and two items were loaded $< 0.5$; 0.453 and 0.449 for items 23 and 38, respectively. Removing the item improved the model fit. For the study population, item to factor correlation analysis showed that the modified version of HBM with 24 items was a valid model.

The maximum likelihood ratio was used to estimate model fitness and the $\chi2$ test was used as a measure of fit between the sample covariance and fitted covariance matrices, $\chi2 = 472.336$, df = 284 (p > 0.05), normed $\chi2$ (472.336/284) = 1.663 which is $\leq 3$. The GFI was 0.839, and AGFI 0.803, both of which were acceptable $\geq 0.80$.

In addition to the $X^2$ test, other fit indices were used to evaluate model fitness, including the CFI = 0.927 and NFI = 0.833 ($\geq 0.90$), RMR = 0.060 ($\leq 0.08$) and RMSEA = 0.059 ($\leq 0.06$).

These data all demonstrate an acceptable model fit, except for the NIF. All indices were within limits and with significant fit. Generally, tests of the goodness-of-fit of the model were conducted, as summarized by Gaskin, J. & Lim, J. (2016), in the, "Model Fit Measures," AMOS Plugin, and indicated an excellent model fit [29] in Table 4 and Fig 1.

**Table 4. Goodness fit indices of the modified HBM for cervical cancer and VIA among health care professionals, Ethiopia.**

| Measure | Estimate | Threshold | Interpretation |
|---------|----------|-----------|----------------|
| CMIN | 396.584 | — | — |
| DF | 236.000 | — | — |
| CMIN/DF | 1.680 | Between 1 and 3 | Excellent |
| CFI | 0.937 | >0.95 | Acceptable |
| SRMR | 0.059 | <0.08 | Excellent |
| RMSEA | 0.059 | <0.06 | Excellent |
| PClose | 0.067 | >0.05 | Excellent |

Note: **CMIN:** Chi-square; **DF**: Degree of freedom; **CFI:** comparative fit index; **SRMR:** standardized root mean square; **RMSEA:** root mean square error of approximation

## Discussion

The HBM has been used in many studies to create measurement tools to identify factors that influence health behaviours across various health conditions and populations. In this study, we evaluated the validity and reliability of a modified HBM for cervical cancer and screening using VIA among female healthcare professionals employed in a large tertiary academic health sciences center. CFA, found that 24 items loaded on six factors. The results revealed that this instrument is suitable to evaluate beliefs regarding cervical cancer screening for a sample of Ethiopian healthcare professionals. Other studies have evaluated the psychometric properties of HBM-related measures associated with HPV vaccination and screening using either exploratory or confirmatory factor analyses to evaluate the factor structures underlying HBM [20]. Most studies retained between 4 and 5 factors which is very close to the findings of the present study in which 6 factors were retained and discrepant with one other study, which yielded a 10-factor structure [30].

The revision of the original 42-item instrument presented here resulted in a shorter yet equally comprehensive basis for measuring HBM constructs. Although the initial 42-item instrument had adequate psychometric properties, the conciseness of the revised version translates into a reduced respondent burden, which may help to enhance survey completion rates in clinical settings. In this case, the instrument was reduced by 18-items (42.9%) with apparently minimal loss of information, if any; however, validation of the reduced instrument in different samples is still necessary and should be the focus of future work.

A few studies have used either exploratory or confirmatory factor analyses to evaluate the factor structures underlying HBM. The subscale reliabilities for these studies varied considerably. Cronbach's alpha ranged from 0.55 in Marlow et al. (2009) to 0.96 in Kahn et al. (2008) and the value for all subscales in this study was 0.806 which is within this range [31, 32]. The internal consistency and reliability of the shortened version of the scale remained strong. The lowest alpha value (0.7) was for susceptibility to cervical cancer, which had only three items. Other psychometric studies of health motivation sub scales have also reported lower reliability using 3-items [20]. The number of items in a subscale has an impact on Cronbach's alpha. Thus, the fact that the lowest alpha was > 0.70 for a subscale with three items is promising, as lower item burden is associated with higher participation compliance. Thus, our data suggest that the reliability of the revised HBM scale for cervical cancer and VIA is acceptable and warrant further testing in other samples of Ethiopian women.

Based on the results of EFA, all questions were clustered in the subscales: benefit of VIA, perceived seriousness of cervical cancer, barrier to fear of negative outcome and health delivery, self-efficacy, susceptibility to cervical cancer and barriers to health delivery subscales. All subscale questions had an acceptable load factor, separately loaded on the related factor. For example, all

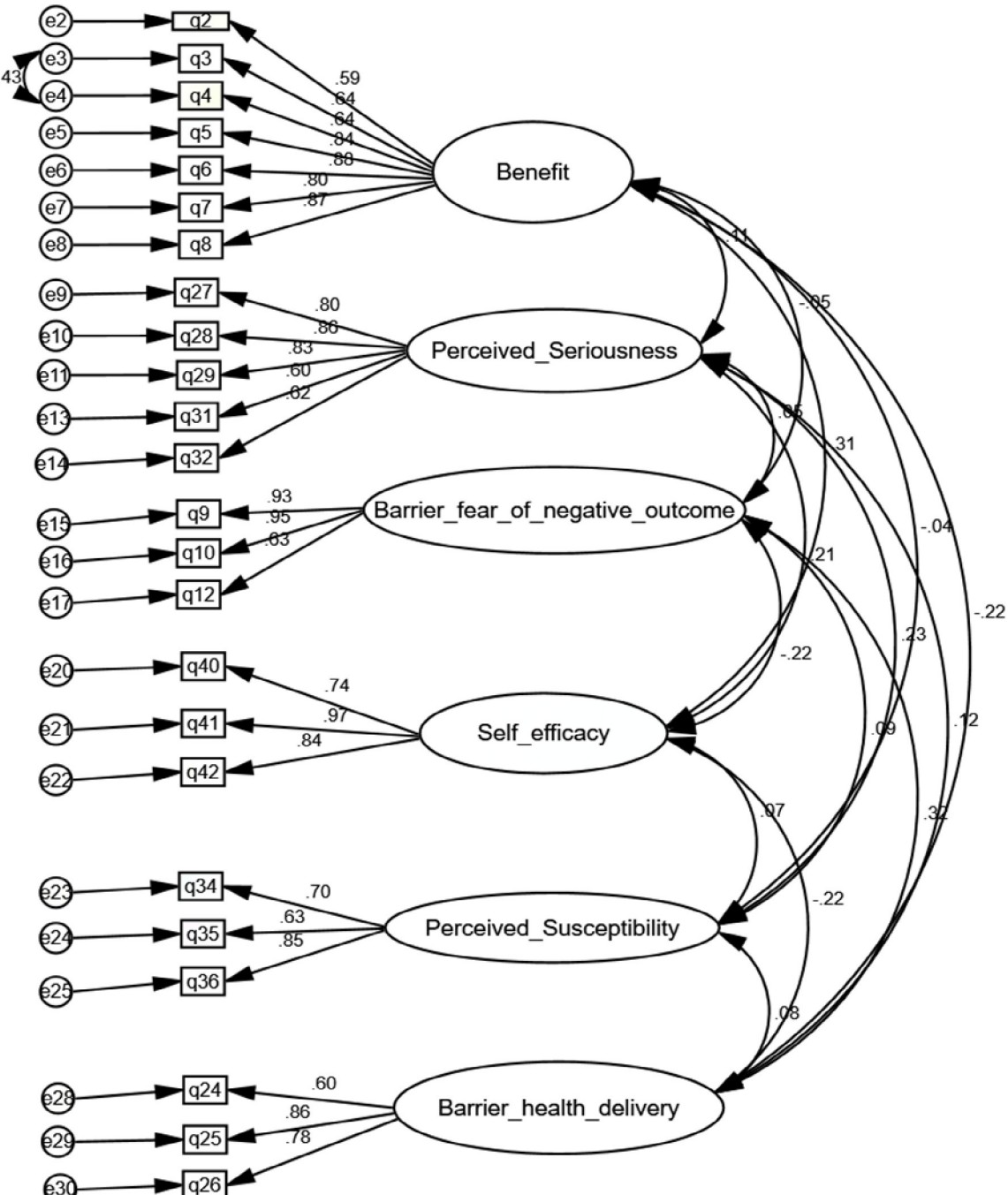

**Fig 1. Six-factor structure of modified HBM for cervical cancer and VIA among health care professionals, Ethiopia.**

questions of the self-efficacy scale were loaded on one factor, similar to a method used by authors of a study on the Iranian version of Champion's Revised Health Belief Model Scale for Breast Cancer screening [33]. Decision-making and health behaviour could be influenced by fear theory where it interacts with other components of the HBM to promote healthy behaviour.

According to the results of EFA, three items of the health motivation subscale, namely; question 37, "I eat well-balanced meals for my health; question 38, "I exercise at least three

times a week for my health; and question 39, "I have regular health check-ups even when I am not sick, question 39". Question 37 was unsatisfactory among healthcare providers, consistent with the results of a study of Iranian female students. The remaining two questions were loaded to the self-efficacy factor. Contrary to some previous findings, all items related to the health motivation subscale were loaded on one factor [33].

Under the construct for barrier variables like 'It is difficult to get an appointment for VIA'; so, creating modalities to have short waiting list in the clinics could be a solution to improve screening practice in the clinical area and enhance early detection to minimize cancer related deaths. As the same time this finding implies for researchers to work on the feasibility of the different screening techniques for cervical cancer like self-testing kits and its efficacy as our finding revealed 'I would be ashamed to lie on a gynecologic examination table and show my private parts to have a VIA' researchers may be interested to look in the feasibility, acceptability of the different screening techniques to optimize screening utilization in the general population the case of Ethiopian women.

Although, there was a difference from the original HBM 42 in the number of items and factor structure, CFA was conducted to assess whether the six -factor EFA measurement was appropriate for the study population. CFA revealed that each item had acceptable loading with its factor. All factors had a good inter-factor correlation and measured similar concepts with acceptable model fit indices.

## Study strengths and limitation

A strength of the study was the ability to validate a culturally appropriate modified version of the cervical cancer screening and VIA HBM tool that may be used for future studies among Ethiopian women. There were also several limitations of this study. Due to of the cross-sectional nature of the data used, the stability of the initial and revised dimension scores over time were not examined. Further, data for the present instrument revision were provided by a sample from a specific population, healthcare professionals; hence, the generalizability of the HBM instrument should be examined in other populations of women in Ethiopia, such as those from rural areas and women who are less educated.

## Conclusion and recommendations

In conclusion, the revision of the 42-item adapted version of the HBM for VIA instrument provided a validated and more parsimonious short- form instrument (24-items) that, in the available sample of health care professionals, appeared to retain the psychometric properties of the original instrument. Directions for future research include testing the instrument in other populations of Ethiopian women. In addition, future studies may consider adapting the modified HBM scale for VIA to examine perception towards cervical cancer and possible barriers that influences screening behaviours. Assessing the perceptions of women about screening and perhaps helping to evaluate interventions to increase screening in LMICs, such as Ethiopia. Future studies comparing the original 42-item and updated 24 items tool with external measures to support external validity of each construct and subscale will be beneficial and provide greater understanding of the psychometric properties of the tool.

## Supporting information

**S1 Checklist. Inclusivity in global research.**
(DOCX)

**S2 Checklist. STROBE statement—Checklist of items that should be included in reports of observational studies.**
(DOCX)

**S1 File.**
(DOCX)

**S2 File.**
(PDF)

## Acknowledgments

The authors would like to thank the individuals who chose to participate in this study and Dr. Deborah Bruner, Emory University, for reviewing the manuscript.

## Author Contributions

**Conceptualization:** Semarya Berhe Lemlem, Rebecca A. Gary, Katherine A. Yeager, Mitike Molla Sisay, Melinda K. Higgins.

**Data curation:** Semarya Berhe Lemlem, Rebecca A. Gary, Katherine A. Yeager, Mitike Molla Sisay, Melinda K. Higgins.

**Formal analysis:** Melinda K. Higgins.

**Methodology:** Semarya Berhe Lemlem, Rebecca A. Gary, Katherine A. Yeager, Mitike Molla Sisay, Melinda K. Higgins.

**Validation:** Rebecca A. Gary, Mitike Molla Sisay, Melinda K. Higgins.

**Writing – original draft:** Semarya Berhe Lemlem.

**Writing – review & editing:** Semarya Berhe Lemlem, Rebecca A. Gary, Katherine A. Yeager, Mitike Molla Sisay, Melinda K. Higgins.

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
