## [Decision Letter · Decision Letter 0]

18 Apr 2023

PONE-D-23-01517Psychometric properties of a modified health belief model for cervical cancer and visual inspection with acetic acid among healthcare professionals in EthiopiaPLOS ONE

Dear Dr. Lemlem,

Thank you for submitting your manuscript to PLOS ONE. After careful consideration, we feel that it has merit but does not fully meet PLOS ONE’s publication criteria as it currently stands. Therefore, we invite you to submit a revised version of the manuscript that addresses the points raised during the review process.

We look forward to receiving your revised manuscript.

Kind regards,

Md Mohsan Khudri

Academic Editor

PLOS ONE

Journal Requirements:

- 10.1016/j.apnr.2016.09.004 YAPNR 50841

In your revision ensure you cite all your sources (including your own works), and quote or rephrase any duplicated text outside the methods section. Further consideration is dependent on these concerns being addressed.

Important: If there are ethical or legal restrictions to sharing your data publicly, please explain these restrictions in detail. Please see our guidelines for more information on what we consider unacceptable restrictions to publicly sharing data: http://journals.plos.org/plosone/s/data-availability#loc-unacceptable-data-access-restrictions  Note that it is not acceptable for the authors to be the sole named individuals responsible for ensuring data access.

Additional Editor Comments :

We have now received two reviewers’ reports for your manuscript. In addition to some other issues, reviewers asked several queries regarding the study’s sampling mechanisms and the reliability and validity of the scale, which are all valid concerns. The reviewers’ points are clearly laid out in their reviews which are appended below.

Reviewers' comments:

Reviewer's Responses to Questions

**Comments to the Author**

1. Is the manuscript technically sound, and do the data support the conclusions?

Reviewer #1: No

Reviewer #2: No

2. Has the statistical analysis been performed appropriately and rigorously? 

Reviewer #1: Yes

Reviewer #2: No

3. Have the authors made all data underlying the findings in their manuscript fully available?

Reviewer #1: Yes

Reviewer #2: Yes

4. Is the manuscript presented in an intelligible fashion and written in standard English?

Reviewer #1: No

Reviewer #2: Yes

5. Review Comments to the Author

Reviewer #1: This manuscript could be an important reference for future studies. However, minor is still needed to improve the quality of this paper. Please revise the manuscript to address the expressed concerns. After thorough review, I am recommending some revisions. In this regard, kindly address the following comments and suggestions to further improve your manuscript

a. Please write the sampling strategy and date and country of study in abstract

b. It was better if you wrote some of main finding as quantitative or mean ±SD within the abstract

c. The introduction section is very long. You could summarize this section a bit more for readers. Write about the problems, the novelty of your study, and your study goals within the introduction

d. Write about the patients sampling method. Discuss more about your sampling strategy? Did you have sampling frame? how did you access to this frame

e. Where have you collected samples? Write the year and the name of place in which you had done this survey. Furthermore, write about all applied exclusion and inclusion criteria a bit more clearly by which you selected samples for this survey.

f. Explain Face and content validity methods in the methodology section

g. Also explain the methods used for reliability in the methodology section

h. The sample size for exploratory factor analysis should be different from the confirmatory factor validation sample size? It is not possible to do both with one sample

i. What are the data extract’s center characteristics? is it governmental or private, is it referral or not referral and so on, discuss more about it

j. In the discussion section I would like to see a more profound discussion about the findings

Reviewer #2: Dear editor in journal of PLOS ONE

Thank you for your invitation to review manuscript entitled "psychometric properties of a modified health belief model for cervical cancer and visual inspection with acetic acid among healthcare professionals in Ethiopia"

Introduction

Comment 1:

Please provide more information fear theories regarding behavior change.

Comment 2:

Please added some sentence regarding constructs of HBM such as cues to action in which missing in your study.

Sample size

Comment 1:

Please clarify, how did you recruited sample of study.

Comment 2:

Did sample of study enough knowledge about the symptoms of cervical cancer?

Statically analysis

Comment 1:

Usually, exploratory and confirmatory factor analysis is performed on separate samples. Did the authors do both on 194 people and on separate samples?

Method

Comment 1:

Please added some sentences regarding visual inspection with acetic acid (VIA).

Discussion

Please improve discussion section with applied fears theories to prevention behavior.

6. PLOS authors have the option to publish the peer review history of their article (what does this mean?). If published, this will include your full peer review and any attached files.

Reviewer #1: No

Reviewer #2: No

---

## [Author Response · Author response to Decision Letter 0]

5 Jul 2023

PONE-D-23-01517

Psychometric properties of a modified health belief model for cervical cancer and visual inspection with acetic acid among healthcare professionals in Ethiopia

First and foremost, we are grateful for PLOS ONE academic editors and reviewers for the insightful inputs provided to improve the quality of our work. 

Response to editors

Qs.1. Please ensure that your manuscript meets PLOS ONE's style requirements, including those for file naming.

Response: After a thorough review of PLOS ONE style template, we have resubmitted the manuscript as per the requirement of PLOS ONE. 

Qs.2. Please upload a completed version of your questionnaire as Supporting Information when you resubmit your manuscript.

Response: The final version of the completed questionnaire has been uploaded. 

Qs.3. We noticed you have some minor occurrence of overlapping text with the following previous publication(s), which needs to be addressed: 10.1016/j.apnr.2016.09.004 YAPNR 50841

Response: The exact URL specified in this comment was not traceable and some of similar articles were about dental hygiene and so forth and this doesn’t have any correlation to our study; but through review of citation in reference has been made extensively. 

Qs.4. Grant information provided in the ‘Funding Information’ and ‘Financial Disclosure’ sections do not match.

Response: The correct grant numbers for the awards received for our study is RD/LT-159/2019 by the office of the Director for Research in collaboration with SIDA Grant coordination called AAU women researchers; to match this we have attached the letter as a supplement file and addressed the correction to match with the ‘Funding Information’ section. 

Qs.4. Data Availability statement, you have not specified where the minimal data set underlying the results described in your manuscript can be found. PLOS defines a study's minimal data set as the underlying data used to reach the conclusions drawn in the manuscript and any additional data required to replicate the reported study findings in their entirety. All PLOS journals require that the minimal data set be made fully available

Response: to fulfil the requirement by PLOS.the data set used for this study could be traced with the following DOI number: https://doi.org/10.20372/aau_rdr/GXJ5UR

Inclusivity in global research

Response: The checklist has been filled out and has been specified to inclusivity in global research has been uploaded as a supporting Information (S39 Checklist). And a subsection has been added on the method section of the manuscript.

Response to reviewer #1: 

a. Please write the sampling strategy and date and country of study in abstract

Response: Thank you so much and as per the comment provided, we have addressed the concern in the abstract section and we have added the sampling strategy, study period and study setting which was collected in 2020 in Addis Ababa, Ethiopia

b. It was better if you wrote some of main finding as quantitative or mean ±SD within the abstract

Response: Number of participants and findings from the socio-demographic characters has been explained with mean and SD like age of participants. 

c. The introduction section is very long. You could summarize this section a bit more for readers. Write about the problems, the novelty of your study, and your study goals within the introduction

Response: Really the points raised is very crucial and we accepted and have written the introduction section as per your suggestion in the revised manuscript/ summarized and precise. 

d. Write about the patients sampling method. Discuss more about your sampling strategy? Did you have sampling frame? how did you access to this frame

Response: Thank you very much for the raised point and we have addressed the sampling strategy that we have followed to come up with these sample. The sampling frame for this study was based on participants’ profile obtained from the office of human resource of the CHS, AAU. Samples were proportionally allocated based on the number of female staffs available in the units/ departments. Health care professionals who fulfil the inclusion criteria and willing to participate were approached face to face to fill out the semi structured questionnaires. 

e. Where have you collected samples? Write the year and the name of place in which you had done this survey. Furthermore, write about all applied exclusion and inclusion criteria a bit more clearly by which you selected samples for this survey.

Response: Study participants were professionals working at college of health sciences (CHS) of Addis Ababa University. The college comprised of 4 schools and 1 teaching hospital. The survey was conducted at college of health sciences; the college has teaching hospital named Tikur Anbessa Specialized Hospital (TASH); which is the largest hospital in Ethiopia. Inclusion criteria was being female healthcare professionals aged 21- 65 years, no prior history of cervical cancer, speaks English and employed full-time. For this survey students and non-health care professionals were excluded from the study. 

f. Explain Face and content validity methods in the methodology section

Response: A key property of validity is the extent to which a measure captures what it is intended to measure and it’s important to get an indicator of face validity at an early stage in the research process or whenever you’re applying an existing test with different populations; as a result. Face and content validity was checked by health care professionals working in different site from the actual data collection site, senior professionals and researchers and feedback was suggestive of good validity.

g. Also explain the methods used for reliability in the methodology section

Response: Thank you and the reliability the internal consistency of the data set was checked by Cronbach alpha value and as per the suggestion you provided this text description: the method used to check the reliability has been explained in the manuscript. 

h. The sample size for exploratory factor analysis should be different from the confirmatory factor validation sample size? It is not possible to do both with one sample

Response: We are really very grateful for the crucial comment you have provided regarding sample size to use for EFA and CFA. When evaluating the psychometric properties of an assessment, researchers can perform an exploratory factor analysis (EFA), followed by a confirmatory factor analysis (CFA) on the same dataset (the whole-sample strategy) to evaluate the model structure. However, the model structure obtained by the whole-sample strategy is based on only one dataset and is, therefore, subject to capitalization on chance or be prone for overfitting of the model. As you have commented to strengthen the generalizability of models, researchers suggest conducting cross-validation and applying different datasets in practice. However; we failed to do that mainly with feasibility concern to collect multiple data set and not having large samples to conduct the split data strategy. With the reason provided above we conducted EFA and CFA on the same data set; and if this is considerable with limitation. And if this is not possible by any means we would like to kindly consider reporting the finding of EFA only for the submitted manuscript and maintain the rigor of our study.

i. What are the data extract’s center characteristics? is it governmental or private, is it referral or not referral and so on, discuss more about it

Response: Tikur Anbessa Hospital is government owned referral hospital and it is an institution where specialized clinical services that are not available in other public or private institutions are rendered to the whole nation. 

j. In the discussion section I would like to see a more profound discussion about the findings

Response: 

Response to reviewer #2: 

Introduction

Comment 1:

Qs; Please provide more information fear theories regarding behaviour change.

Response: Fear theory is a psychological concept that explains the emotional and physiological response to perceived threats. The HBM is psychological model that attempts to explain and predict health behaviours by focusing on individual’s attitude and beliefs. And the details of the HBM is explained well within the manuscript 

Comment 2:

Qs; Please added some sentence regarding constructs of HBM such as cues to action in which missing in your study.

Response: Cues to action of the HBM posits that a cue, or trigger, is necessary for prompting engagement in health-promoting behaviours. Cues to action can be internal or external. With the perception and their scope practice could triggers as the study was among health care professionals’ details was not indicated on cues. 

Sample size

Comment 1:

Qs; Please clarify, how did you recruit sample of study.

Response: Thank you very much for the raised point and we have addressed the sampling strategy that we have followed to come up with these sample. The sampling frame for this study was based on participants’ profile obtained from the office of human resource of the CHS, AAU. Samples were proportionally allocated based on the number of female staffs available in the units/ departments. Health care professionals who fulfil the inclusion criteria and willing to participate were approached face to face to fill out the semi structured questionnaires. Comment 2:

Qs; Did sample of study have enough knowledge about the symptoms of cervical cancer?

Response: Knowledge about cervical cancer was assessed using 28 items to measure; risk factors, symptoms, preventive measures, and its screening. Among the 28 items used; some items have multiple responses, hence we scored ‘1’ for each correct response and ‘0’ for the wrong responses. Health care professionals in this study had a mean knowledge of 15.63, +3.14, out of a possible range of 0 to 28. 

Statically analysis

Comment 1:

Usually, exploratory and confirmatory factor analysis is performed on separate samples. Did the authors do both on 194 people and on separate samples?

Response: We are really very grateful for the crucial comment you have provided regarding sample size to use for EFA and CFA. When evaluating the psychometric properties of an assessment, researchers can perform an exploratory factor analysis (EFA), followed by a confirmatory factor analysis (CFA) on the same dataset (the whole-sample strategy) to evaluate the model structure. However, the model structure obtained by the whole-sample strategy is based on only one dataset and is, therefore, subject to capitalization on chance or be prone for overfitting of the model. As you have commented to strengthen the generalizability of models, researchers suggest conducting cross-validation and applying different datasets in practice. However; we failed to do that mainly with feasibility concern to collect multiple data set and not having large samples to conduct the split data strategy. With the reason provided above we conducted EFA and CFA on the same data set; and if this is considerable with limitation. And if this is not possible by any means we would like to kindly consider reporting the finding of EFA only for the submitted manuscript and maintain the rigor of our study.

Method

Comment 1:

Please add some sentences regarding visual inspection with acetic acid (VIA).

Response: Thank you and description has been provided in the introduction section prior submission and we have also added some sentence about VIA in the method section as well which stated as Visual inspection with acetic acid (VIA) is visualization of woman’s cervix to detect precursors of cervical cancer after application of acetic acid (ordinary table vinegar) on her cervix and it is simple, low-cost, and efficient alternative to cytologic testing in low-resource areas. 

Discussion

Please improve discussion section with applied fears theories to prevention behaviour.

Response: Fear theory works within two primary constructs of the HBM; perceived susceptibility and perceived severity to determine the level of fear and influence the likelihood of engaging in health promoting behaviours. Fear alone may not be sufficient to promote behaviour change as other factors play pivotal role in shaping health behaviour. Some points have been incorporated in the discussion section. And some supporting points for implementation to practice has been explained in this section to profoundly indicate the relevance of this study outcome in the process of optimising cervical screening utilization.

---

## [Decision Letter · Decision Letter 1]

4 Dec 2023

Psychometric properties of a modified health belief model for cervical cancer and visual inspection with acetic acid among healthcare professionals in Ethiopia

PONE-D-23-01517R1

Dear Dr. Lemlem,

We’re pleased to inform you that your manuscript has been judged scientifically suitable for publication and will be formally accepted for publication once it meets all outstanding technical requirements.

Kind regards,

Md Mohsan Khudri

Academic Editor

PLOS ONE

Additional Editor Comments (optional):

Dear Dr. Lemlem,

We’re pleased to inform you that your manuscript has been judged scientifically suitable for publication and will be formally accepted for publication once it meets all outstanding technical requirements.

Within one week, you’ll receive an e-mail detailing the required amendments. When these have been addressed, you’ll receive a formal acceptance letter, and your manuscript will be scheduled for publication.

Reviewers' comments:

Reviewer's Responses to Questions

**Comments to the Author**

1. If the authors have adequately addressed your comments raised in a previous round of review and you feel that this manuscript is now acceptable for publication, you may indicate that here to bypass the “Comments to the Author” section, enter your conflict of interest statement in the “Confidential to Editor” section, and submit your "Accept" recommendation.

Reviewer #1: (No Response)

Reviewer #3: All comments have been addressed

2. Is the manuscript technically sound, and do the data support the conclusions?

Reviewer #1: (No Response)

Reviewer #3: (No Response)

3. Has the statistical analysis been performed appropriately and rigorously? 

Reviewer #1: (No Response)

Reviewer #3: (No Response)

4. Have the authors made all data underlying the findings in their manuscript fully available?

Reviewer #1: (No Response)

Reviewer #3: (No Response)

5. Is the manuscript presented in an intelligible fashion and written in standard English?

Reviewer #1: (No Response)

Reviewer #3: (No Response)

6. Review Comments to the Author

Reviewer #1: Although the authors responded to reviewer comments and addressed the issues identified, I believe that the manuscript not suitable for publication in Plose One. While I recognize the decision of "reject" may be disappointing for the authors, I do think this research has important weaknesses in the method section.

Reviewer #3: This manuscript was very interesting, clear writing and understandable and the authors' ideas was clearly expressed

7. PLOS authors have the option to publish the peer review history of their article (what does this mean?). If published, this will include your full peer review and any attached files.

Reviewer #1: No

Reviewer #3: No

---

## [Editor Report · Acceptance letter]

11 Dec 2023

PONE-D-23-01517R1 

Psychometric properties of a modified health belief model for cervical cancer and visual inspection with acetic acid among healthcare professionals in Ethiopia 

Dear Dr. Lemlem:

I'm pleased to inform you that your manuscript has been deemed suitable for publication in PLOS ONE. Congratulations! Your manuscript is now with our production department. 

Kind regards, 

on behalf of

Dr. Md Mohsan Khudri 

Academic Editor

PLOS ONE